# Regulatory Mechanisms of Heat Stress Response and Thermomorphogenesis in Plants

**DOI:** 10.3390/plants11243410

**Published:** 2022-12-07

**Authors:** Yunzhuan Zhou, Fuxiang Xu, Yanan Shao, Junna He

**Affiliations:** Beijing Key Laboratory of Development and Quality Control of Ornamental Crops, College of Horticulture, China Agricultural University, Beijing 100193, China

**Keywords:** heat stress, heat shock transcription factor, heat shock protein, phytochrome interacting factor 4, thermomorphogenesis

## Abstract

As worldwide warming intensifies, the average temperature of the earth continues to increase. Temperature is a key factor for the growth and development of all organisms and governs the distribution and seasonal behavior of plants. High temperatures lead to various biochemical, physiological, and morphological changes in plants and threaten plant productivity. As sessile organisms, plants are subjected to various hostile environmental factors and forced to change their cellular state and morphological architecture to successfully deal with the damage they suffer. Therefore, plants have evolved multiple strategies to cope with an abnormal rise in temperature. There are two main mechanisms by which plants respond to elevated environmental temperatures. One is the heat stress response, which is activated under extremely high temperatures; the other is the thermomorphogenesis response, which is activated under moderately elevated temperatures, below the heat-stress range. In this review, we summarize recent progress in the study of these two important heat-responsive molecular regulatory pathways mediated, respectively, by the Heat Shock Transcription Factor (HSF)–Heat Shock Protein (HSP) pathway and PHYTOCHROME INTER-ACTING FACTOR 4 (PIF4) pathways in plants and elucidate the regulatory mechanisms of the genes involved in these pathways to provide comprehensive data for researchers studying the heat response. We also discuss future perspectives in this field.

## 1. Introduction

A rise of 2.5–5.4 °C in the global surface temperature during the 21st century has been projected by the Intergovernmental Panel on Climate Change [1]. Plants are responsive to temperature changes, some species even to differences of 1 °C [2]. A characteristic set of cellular and metabolic responses occur when plants are exposed to excess heat, with temperatures at least 5 °C above their optimal growing conditions [3,4]. Although different plants (crops) have discrete temperature thresholds in response to extreme and moderate high temperature (Table 1), climate change and global warming are severely hindering the normal growth and development of many important crops, threatening food safety [3].

Plants have evolved various complex pathways and mechanisms to respond to elevated temperatures. Therefore, the biology and physiology of the heat response, including the extreme heat stress response (HSR) and the moderately elevated temperature response called thermomorphogenesis, need to be studied at both cellular and organic levels [18,19,20]. Under heat stress conditions, plants exhibit many molecular responses, such as an increase in the synthesis of heat shock proteins and hormones to survive [21], while under moderately elevated temperature, below the heat-stress range, plants change their growth and development morphologically and architecturally, with, e.g., hypocotyl elongation, and leaf hyponasty, and accelerate flowering to adapt to the warm temperatures [20]. Some pivotal molecular factors that promote plant thermomorphogenesis and adaption to heat stress have been gradually identified through the application of forward and reverse genetics [22]. Here, we firstly used the Statistical Analysis Toolkit for Informetrics (SATI, http://sati.liuqiyuan.com (accessed on 3 January 2022) to identify the main keywords in the thermoresponsive field in recent years (2016–2022) by an advanced search with keywords related to heat (TI = “Heat Stress” OR “thermotolerance” OR “high-temperature” OR “high temperature” OR “Thermomorphogenesis” OR “thermoresponsive” OR “Heat Shock”) in the core collection of Web of Science and limited the subject terms to plants and the time scale to the last 6 years (2016–2022) [23]. A total of 3, 414 articles were searched, and the top keyword distribution was analyzed (Figure 1).

The occurrence frequency of keywords in these articles revealed that RNA-seq, proteomics, metabolomics, mass spectrometry, and QTL methods have all been used in thermoresponsive research in recent years. The main study subjects of thermoresponsive research works are not only the model plant Arabidopsis, but also common and important crops, such as wheat, maize, rice, and soybean. Tomato is the most popular vegetable crop in thermoresponsive studies. Furthermore, based on these keywords, we found that the main damage caused by high temperature at the cellular level in plants regards intracellular proteostasis, DNA, apoptosis, autophagy, and alternative splicing (AS) events of genes. Consequently, these damages lead to macroscopic disruptions in important plant traits, including germination, spikelet fertility, grain quality, and plant yields.

In addition, the distance between the keywords in Figure 1 suggest that HSPs, which are directly targeted by HSFs, and some other molecular chaperone proteins have a specific role in proteostasis regulation in plant cells under high temperature, as confirmed by many studies. In addition to the important role of HSPs and molecular chaperones in the regulation of plant cell proteostasis, recent studies have also shown that autophagosome-dominated autophagy pathways in plant cells play a very important role in regulating changes in proteostasis caused by heat. Phytohormones, especially ABA and SA, are crucial not only for the normal growth and development of plants, but also for thermoresponsive regulation. Another molecular regulatory mechanism that has received attention with respect to the heat response is the ROS pathway, which plays a crucial role as a second messenger in the heat response to transduce the temperature signal quickly. According to a summary of recent studies, there is a close connection between the heat stress response and drought stress as well as the photosynthesis pathway, which also indicates the diversity and complexity of plant thermoresponsive molecular regulatory mechanisms.

Here, based on the keywords distribution and abundant literature available on how plants tolerate extreme heat stress and moderately increased temperatures, we review two main molecular regulatory mechanisms activated in plants in response to elevated temperatures, i.e., the heat stress response and the thermomorphogenesis response, which are mediated by the Heat Shock Transcription Factor (HSF)–Heat Shock Protein (HSP) pathway and the PHYTOCHROME-INTERACTING FACTOR 4 (PIF4) pathway, respectively.

## 2. HSFs and HSPs-Mediated Heat Stress Responses (HSR) in Plants

Although the thermal regulatory mechanisms may have diverged in part in different species, most of the thermal regulatory mechanisms in Arabidopsis—which has served as the prototype for the mechanistic exploration of HS in higher plants—can conceivably be extended to crops such as wheat [12,24,25], rice [26,27,28], maize [29,30,31], tomato [15,32], lily [14,33,34,35,36], particularly the HSF–HSP pathway [21,37]. Under heat stress (Figure 2), heat signaling is transmitted, through the second messengers ROS and calcium ions, to HSFA1, which is the master regulator of the HSF–HSP pathway [13,37,38,39,40,41]. The released HSFA1 further activates the expression of the *HSFA2* and *HSFA7* genes [42,43,44]. *HSFA2* is the main enhancer of the HSR in this pathway [45,46]. It directly activates the expression of the downstream HSP genes and, more importantly, interacts with HSFA1 and HSFB1, forming a complex that further enhances the expression of downstream heat response genes [47]. As a master gene of the HSF–HSP pathway, HSFA1 can also directly induce the expression of the drought-response gene *DREB2A*. DREB2A is a downstream regulator of both osmotic and heat stress and positively controls osmotic- and heat stress-inducible gene expression; especially, it promotes the expression of the *HSFA3* gene to enhance plant heat tolerance [48,49]. Simultaneously, HSFA1 can regulate *HSFA3* expression by negatively regulating the expression of *HSFA6B*, which directly induces *DREB2A* [5]. *HSFA2* induces the production of heat stress proteins, such as HEAT SHOCK BINDING PROTEIN 2 (HSBP2), as well as of catalytic metabolism-related enzymes, such as galactinol synthase 2 (GOLS2) and raffinose synthases (RAFS), which are two key enzymes in raffinose biosynthesis [50]. Consequently, raffinose synthesis is promoted, and the heat tolerance of plants is improved [29,51]. More importantly, HSFA2 is required not only for the induction of HS genes but also for HS memory maintenance [37,52,53]. The HSFA2 protein interacts with heat shock protein HSP90.1, which is an important regulatory factor of thermotolerance [54,55]. Meanwhile, HSP90.1 can also form a complex with ROF1/AFKBP62 (rotamase FKBP 1) to respond to heat stress [55,56]. Under normal temperature conditions, the HSP90.1–ROF1 complex is located in the cytoplasm, but following heat stress, HSFA2 interacts with HSP90.1–ROF1, forming an HSFA2–HSP90.1–ROF1 complex which enters the nucleus. This HSFA2–HSP90.1–ROF1 complex is thought to be essential to promote the transcriptional activity of HSFA2 and facilitate the synthesis of HSPs during heat stress recovery, making the plant more strongly responsive to an upcoming recurrence of heat stress [55,56]. However, NBR1, which is a plant homolog of the mammalian autophagic cargo receptor SQSTM1/p62, hinders HSFA2-mediated heat stress memory by degrading the HSP90 and ROF1 proteins through the autophagic pathway, reducing their interactions and further affecting their interactions with HSFA2 [57].

The production of small amounts of ROS also directly induces the expression of *HSFA1* and *HSFA4*, which activate the downstream HSR genes [5,28,32]. However, HSFA5 can interact with HSFA4, inhibiting the transcriptional activity of HSFA4 involving the downstream heat-responsive genes [58]. Interestingly, an excessive increase in ROS content due to increased intracellular misfolding of proteins under heat stress can result in cell death and activate the autophagy pathway [9,59]. Autophagy shows an important role not only in plant development but also in responses to abiotic stresses and biotic stresses [60,61,62,63]. The autophagy-related protein ATG18 promotes the increase of catalase (CAT), peroxidase (POD), and superoxide dismutase (SOD) in the ROS scavenging system through the autophagy pathway to reduce the excess ROS, allowing a correct downstream transmission of heat stress signals [9].

Simultaneously, an increase in protein misfolding events can also induce the expression of *INOSITOL-REQUIRING ENZYME 1* (*IRE1*), an RNA splicing factor localized in the endoplasmic reticulum membrane [64,65]. Furthermore, IRE1 triggers the unfolded protein response (UPR) within the endoplasmic reticulum (ER) [30,66]. The thermal activation of IRE1 can induce the alternative splicing (AS) of the bZIP transcription factor family members bZIP60 and bZIP74 mRNA to form a nuclear-imported form of mRNA [64,65,66,67,68,69]. Subsequently, the nuclear-imported form of bZIP mRNA upregulates the HSFA family gene *HSFTF13* (*HSFA6B*), which further activates the expression of the downstream HSP genes, linking of the UPR response and the HSR response in plants [30]. In addition, the major regulatory factors of the HSF–HSP pathway, HSFA1, HSFA2, and HSFB1, can form complexes with HSP70, sHSP, and HSP90, respectively, at a normal temperature, limiting the HSFs levels. Under heat stress, these complexes are separated, thereby increasing the content of HSFs, which further promotes the expression of the downstream HSR genes and other heat-responsive genes, thus contributing to the establishment of heat tolerance [47,70,71,72]. In addition, it was found that HSP90 could bind to high expression of osmotically responsive gene 1 (HOS1) under high temperature, stabilizing the protein activity of HOS1, which promotes the expression of RECQ2 DNA helicase. The high expression of *RECQ2* allowed a further repair of the extent of DNA damage caused by high temperatures to improve heat tolerance [73].

Conversely, heat stress also induces negative regulatory mechanisms for the establishment of thermotolerance. Heat stress increases the post-translational modification (PTM) of genes, which inhibits the protein activity of HSFs at high temperatures through chaperone-mediated ubiquitination and degradation by the ubiquitin proteasome system (UPS) and negatively regulates cellular HSFs network activity [70]. It is well documented that alternative splicing (AS) is an important post-transcriptional regulatory mechanism involved in plant responses to temperature variation [74,75,76]. Some studies indicated that temperature variations induce alternative splicing of *HSFA2*, and different spliced forms of HSFA2 possess differential protein localization and activity. The efficiency of AS may be an important regulatory mechanism in thermotolerance [27,77]. Moreover, some studies showed that the activity of HSFs can also be regulated by heat shock binding protein (HSBP), which interferes with their oligomerization. HSFBP2, an HSF-binding protein in Arabidopsis induced by HSFA2, can reduce the DNA binding capacity of HSFA2 and functions as a negative regulator of the HSR [29,78,79].

To date, the conservation of the HSF–HSP pathway in Arabidopsis and other species has been well elucidated. The significantly conserved thermal functions of some key HSR genes in crops have also been revealed, such as those of HSFA1b in wheat [25], OsbZIP58 in rice [27], HSFA2, HSBP2, and bZIP60 in maize [29,30,31], HSP40 and HSFs in tomato [15,32], and HSFAs in lily [14,33,34,35,36]. However, heat tolerance research in crops is only about cloning, and the thermal functions of key HSR genes and specific regulatory networks still need further exploration.

## 3. PIF4-Mediated Thermomorphogenesis under Warm Temperatures

### 3.1. PIF4 Promotes Plant Thermomorphogenesis by Regulating Auxin Response

So far, most of the regulatory mechanisms of thermomorphogenesis in plants have been mainly studied and interpreted in the model plant Arabidopsis and have rarely been further extended to other plant species. Therefore, it will be important in the future to uncover whether some Arabidopsis regulatory mechanisms of thermomorphogenesis are conserved in other higher plants, particularly in crops. In the model plant Arabidopsis, moderately elevated temperatures promote thermomorphogenesis, characterized by hypocotyl elongation, leaf hyponasty, and early flowering [20]. Many studies have shown that PHYTOCHROME-INTERACTING FACTOR 4 (PIF4) is a dominant regulator not only in the light signal transduction but also in the thermoresponsive pathway [80,81,82,83,84] (Figure 3). PIF4 can integrate the elevated temperature signal into the auxin response and increases the auxin level, which then mediates thermomorphogenesis [45,85,86]. PIF4 promotes the expression of key IAA biosynthetic genes such as *YUCCA 8* (*YUC8*), *CYTOCHROME P450 FAMILY 79B* (*CYP79B*), and *TRYPTOPHAN AMINOTRANSFERASE OF ARABIDOPSIS 1* (*TAA1*) through temperature-mediated binding to their promoters [87,88]. The high-temperature-mediated induction of TAA1 and CYP79B2 is greatly reduced in *pif4*-*101* mutants, indicating that PIF4 plays an important role in the temperature-dependent regulation of the expression of these genes [87]. An elevated auxin level then promotes the expression of AUXIN/INDOLE-3-ACETIC ACID (Aux/IAA) (for example, *IAA4* and *IAA29*), *SMALL AUXIN UPREGULATED RNA* (*SAUR*) genes, and *ARABIDOPSIS THALIANA HOMEOBOX PROTEIN2* (*AtHB2*) involved in cell elongation, which are also reported to be activated by PIF4 in a temperature-dependent manner [81,85,89]. Aux/IAA genes have been found to negatively regulate the activity of AUXIN REPONSE FACTOR (ARF) transcription factors and attenuate auxin responses [90]. However, increased auxin levels promote auxin binding to the receptor INHIBITOR 1/AUXIN SIGNALING F-BOX (TIR1/AFBs), which interacts with AUX/IAA and subsequently initiates the degradation of AUX/IAAs [91]. The degradation of AUX/IAA then releases ARFs, which activates the expression of *SMALL AUXIN UPREGULATED RNA* (*SAURs*), mainly belonging to the SAUR19–24 and SAUR61–68 subfamilies, to drive elongation growth at warm temperatures [92]. More importantly, studies have confirmed that PIF4 also directly enhances the transcription of *YUCCA8* (*YUC8*) by directly binding to the promoter of *YUCCA8*, thus promoting auxin accumulation, which regulates thermomorphogenesis at high temperatures [87,88]. Moreover, increased IAA levels enhance BR signaling to BZR1, which in turn induces PIF4 expression, thus forming a PIF4–AUXIN–BR–PIF4 positive feedback regulatory network for the heat response [93].

### 3.2. Transcriptional and Posttranscriptional Regulation of PIF4

As a key regulator of plant thermomorphogenesis, the regulation of PIF4 is subjected to various regulatory factors at the transcriptional and posttranslational levels [87,88,94]. EARLY FLOWERING3 (ELF3), a proposed thermosensor, mainly acts as a key factor in the evening complex (EC) of the circadian clock [95]. Recent studies have also shown that ELF3 interacts with PIF4 to suppress the transcriptional activity of PIF4 in an EC-independent manner, which blocks the activation of thermoresponsive genes in PIF4-mediated thermosensory pathways [69,96,97]. However, the latest findings suggest that the inhibitory effect of ELF3 on PIF4 is attenuated through the E3 ligase XBAT31-mediated ELF3 degradation in response to warm temperatures in Arabidopsis [98]. XBAT31 ubiquitinates ELF3 through a direct interaction with ELF3 and B-box protein BBX18, which then degrades ELF3 via the 26S proteasome and therefore promotes the release of PIF4. In addition, since ELF3 acts as a core member of EC, XBAT31-mediated ELF3 degradation also inhibits the negatively transcriptional regulation by the EC protein complex of PIF4. Recently, ELF3 has also been proposed to be a thermosensor with polyglutamine (polyQ) repeats within a predicted prion-like domain (PrD) in Arabidopsis [99]. Under warm temperature conditions, ELF3 also releases the inhibitory effect on PIF4 through a liquid–liquid phase separation.

PIF4 was also reported to physically interact with DELLA proteins, but this interaction inhibits elongation growth by preventing PIF4 binding to the promoters of target genes [100,101,102,103,104,105]. Interestingly, the plant hormone gibberellin (GA) can relieve the growth restraint by mediating the degradation of DELLAs and enhancing PIF4 activity at warm temperatures [100,101]. Consistently, elevated temperatures can rapidly upregulate the expression of the GA biosynthesis genes *AtGA20ox1* and *AtGA3ox1* and then increase the GA levels to release PIF4 from DELLAs in Arabidopsis seedlings [106]. In addition, some new findings indicate that the TEOSINTE BRANCHED 1/CYCLOIDEA/PCF (TCP) transcription factors such as TCP5 and TCP17 promote the activity of PIF4 [107,108,109]. Elevated temperatures can rapidly induce the expression of these TCPs and also increase their protein stability. For example, TCP5 not only physically interacts with PIF4 to enhance PIF4 transcriptional activity but also directly binds to the promoter of *PIF4* to regulate the *PIF4* transcript. Therefore, TCPs may also play important roles in regulating the activity of PIF4 during thermomorphogenesis.

FCA, an RNA-binding protein, has been reported to affect the thermosensory flowering process and promote flowering in response to elevated growth temperatures by incorporating ambient temperature signals through an FLC-independent pathway [110,111]. FCA can also suppress the activity of PIF4 to regulate temperature-mediated flowering and hypocotyl growth [112,113]. It has been shown that FCA can induce the dissociation of PIF4 from the *YUC8* promoter by removing H3K4me2 chromatin marks and attenuates PIF4 activity at elevated temperatures, which further negatively regulates the expression of YUC8 and IAA synthesis [113]. In addition, the region of PIF4 binding can be decreased by H2A.Z-nucleosome incorporation into chromatin mediated by ACTIN-RELATED PROTEIN 6 (ARP6) [114]. However, under elevated temperatures, PIF4 can increase the probability of directly binding to the promoters of its target genes, including YUC8, through disassociation of the histone variant H2A.Z. from the nucleosome [93,114]. Under elevated temperatures, the space occupancy of the histone variant H2A.Z in the nucleosome of PIF4 targets was reduced, which freed the PIF4 binding space in these genes’ locus, enhancing their transcriptional regulation and promoting thermomorphogenesis in plants. Therefore, chromatin remodeling may have a prominent role in thermomorphogenesis [93,113,114,115,116].

It is worth noting that H2A.Z-nucleosomes are also present at the PIF4-binding site in the *Flowering locus T* (*FT*) promoter and hinder PIF4 binding [115]. Similarly, the spatial occupancy of the histone variant H2A.Z in nucleosomes of FT is also decreased by elevated temperatures [45]. Thus, PIF4 can directly and strongly bind to the *FT* promoter, promoting the transcription of *FT* and early flowering under warm temperatures [115]. Importantly, an increasing number of studies have identified the important role of histone deacetylases in the heat response [20,24,117,118]. The acetylation of histones increases the occupancy of the nucleosome space of the H2A.Z variant, inhibiting the heat-responsive transcription of these thermoresponsive genes [20,114,117]. Under elevated temperatures, HDA9 is reported to remove the histone acetylation of thermoresponsive gene loci, including the H2A.Z variant, and therefore increase the transcription of these thermoresponsive genes, promoting thermomorphogenesis and the establishment of heat tolerance [119,120].

### 3.3. Light Signal and Temperature Signal Integration through PIF4

Light is an important environmental factor for plant development. Three main classes of photoreceptors, the red (R) and far-red (FR) light-absorbing phytochromes and the UV-A/blue light-absorbing cryptochromes and phototropins, are involved in light signal transduction to adjust plant growth and development [121,122]. Many studies have found that there is signal integration when plants are responding simultaneously to light and elevated temperatures, and PIF4 is emerging as a cellular “hub” of this integration involving photoreceptors that regulates growth and development [81,82,123,124] (Figure 4).

#### 3.3.1. PIF4 Integrates the Darkness Signal with the Elevated Temperature Signal

The evening complex (EC), consisting of ELF3, ELF4, and LUX ARRYTHMO (LUX), is a core component of the circadian clock and responds to elevated temperatures to control hypocotyl growth through the regulation of PIF4 [125,126] (Figure 4a). The EC peaks at dusk, binds to the *PIF4* promoter via the LUX transcription factor, and represses PIF4 transcription in the early night [95]. However, EC activity, as a night transcriptional repressor, is reduced by warm temperatures [95]. There are two key components of light signaling, i.e., DE-ETIOLATED 1 (DET1) and CONSTITUTIVE PHOTOMORPHOGENESIS 1 (COP1), which can up-regulate the expression of *PIF4* by inactivating ELONGATED HYPOCOTYL 5 (HY5), which is the repressor of PIF4 transcription in darkness in response to elevated temperatures [127]. However, recent studies also confirmed that DET1 and COP1 partly directly upregulate *PIF4* expression and positively stabilize the PIF4 protein to promote elongation growth at elevated temperatures [124]. Meanwhile, studies have further confirmed that HY5 mainly inhibits hypocotyl elongation by competitively binding to the promoters of PIF4 target genes, such as *YUC8*, not by the transcriptional repression of PIF4 [124]. At elevated temperatures, the HY5 protein is degraded, and its activity is also decreased, which reduces HY5 competition with PIF4 and thus activates PIF4 target genes [124,128]. The degradation of HY5 is reported to be mediated through the DET1–COP1-dependent degradation pathway in darkness [129]. As a central repressor of Arabidopsis photomorphogenesis, COP1 is highly active in darkness to target numerous transcription factors (TFs) such as HY5 by working in complex with SUPPRESSOR OF PHYA-105 (SPA) proteins in ubiquitin-mediated protein degradation [130,131]. In addition, the constitutive photomorphogenic 10-damaged DNA binding protein 1-DE-ETIOLATED 1 (COP10-DDB1-DET1, CDD) complex can enhance the activity of COP1-SPA E3 ubiquitin ligases, which further promote the degradation of HY5 [132,133].

#### 3.3.2. PIF4 Integrates the Red Light Signal with the Elevated Temperature Signal

Previous studies found that red light promoted hypocotyl extension at warm temperatures, whereas it repressed hypocotyl elongation at low ambient temperatures, indicating that the red light response was strictly temperature-dependent [134]. The light-activated red/far-red light photoreceptor PHYTOCHROME B (PHYB) has been reported to function as a thermosensor in Arabidopsis [2,135]. Under normal conditions, PHYB targets PIF4 for post-transcriptional degradation, whereas elevated temperatures promote the conversation of phyB from active Pfr to inactive Pr (Figure 4b), therefore releasing the inhibitory effects of phyB on PIF4 protein [115,126,136,137,138,139].

In addition, recent studies have found that short hypocotyl under blue 1 (SHB1) and an MYB transcription factor circadian clock-associated 1 (CCA1, the central oscillator components) have a positive regulatory effect on *PIF4* expression in a red light-dependent manner at warm temperatures [126,140]. In response to both red light and elevated ambient temperature, CCA1 interacts with SHB1 and then promotes PIF4 expression through targeting the *PIF4* promoter, therefore desensitizing the light responses for optimal photomorphogenesis and enhancing plant thermomorphogenesis for better survival under elevated ambient temperatures [140]. Recent research has also found another clock-associated gene-mediated balancing thermoresponsive growth system in plants by modulating the level of ELF4 by CCA1 and REVEILLE5 (RVE5) at warm temperatures [141]. In Arabidopsis, both RVE5 and CCA1 are transcriptional repressors and accumulate at warm temperatures, which can reduce the expression of ELF4 to promote hypocotyl growth. They can bind to the same cis element of the ELF4 promoter, and the transcriptional repression activity of RVE5 is weaker than that of CCA1, while the binding of RVE5 to the ELF4 promoter is enhanced by competing for the same cis element with the stronger transcriptional repressor CCA1 at warm temperatures. Such a fine-tuned mechanism provides reasonable thermoresponsive hypocotyl growth mediated by PIF4 at elevated temperatures.

#### 3.3.3. PIF4 Integrates the Blue Light Signal with the Elevated Temperature Signal

In contrast to the effect of red light on hypocotyl elongation, blue light represses high-temperature-induced hypocotyl elongation through the blue light receptor Cryptochrome 1 (CRY1) [142] (Figure 4c). CRY1 and PIF4 occupy the same promoter regions of *YUC8* and *IAA29*. Moreover, warm temperatures also promoted the association of CRY1 with the *YUC8* and *IAA29* promoters. However, CRY1 could not bind to the DNA fragments present in the promoter region of these genes by itself and needs PIF4 to form a complex through direct interaction, in a blue light-dependent manner [142]. The direct interaction between CRY1 and PIF4 subsequently only represses the transcription activity of PIF4, which then reduces auxin biosynthesis in response to elevated temperatures.

Meanwhile, the interaction with the bHLH transcription factor HFR1 (LONG HYPO-COTYL IN FAR RED1) can also sequester the free PIF4 protein, preventing its binding to DNA and downstream transcriptional regulation [100,123]. HFR1 is a negative regulator of temperature responses under monochromatic blue light and is degraded by COP1. HFR1 accumulates in a CRY1-dependent manner as CRYs suppress the E3 ubiquitin ligase activity of COP1 by forming a complex with SPA1 and COP1 in a blue light-dependent manner [123,143,144]. Therefore, CRY1 can also repress PIF4 by inhibiting the COP1-mediated degradation of HFR1.

These results indicate that plant photoreceptors and ambient temperature can mediate morphogenesis through the same signaling component PIF4.

## 4. Conclusions

The response to and survival of plants at elevated temperatures are complex phenomena. Plants have evolved complex physiological systems and a diverse range of tissues and organs that can sense and integrate information from the environment, adapt to the environment, and optimize their growth state. The induction of classical HSPs (chaperones) through the HSF network and auxin-mediated thermomophogenesis as well as the PIF4 network are clearly important, but even this response is not simple and involves several integrators. Other signaling molecules and complex regulation networks orchestrate the heat response, regulating a range of effectors components, all of which contribute to plant adaptation to warm temperatures and their survival under heat stress.

In this review, we presented recent research advances at all these levels of investigation and focused on the key integrators HSFs and PIF4, which may help to understand more fully the mechanisms of plants’ response to heat. The HSF–HSP pathway has dominant roles in the response to heat stress and reduces the damage to cell components and structures caused by extremely high temperatures, increasing plant heat tolerance. PIF4 is a central integrator of environmental information, especially the light signal and the temperature signal, in plants, constituting a signaling loop to optimize photomorphogenesis and enhance thermomorphogenesis. It will likely be a key node to consider for breeding crops resilient to climate change. However, although some studies have shown that there is a crosstalk among these reported thermosensory pathways, more studies are needed to investigate this complex crosstalk. In particular, there are no reports indicating whether there is a certain direct molecular crosstalk between the HSF–HSP pathway and PIF4. These studies will be important to advance thermosensory research and possibly lead to the generation of new plant varieties with sustainable production. Better understanding the molecular mechanisms of the plant response to elevated temperatures and ultimately applying this knowledge to production practices will help to address food security issues linked to increasing population, higher average temperatures, and larger temperature fluctuations.

## Figures and Tables

**Figure 1 plants-11-03410-f001:**
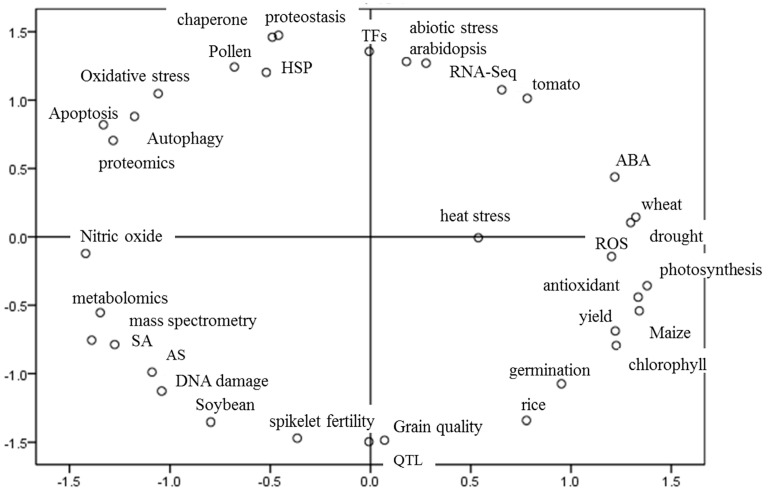
Keywords distribution in heat–related studies in recent years (from 2016 to 2022) by the Statistical Analysis Toolkit for Informetrics.

**Figure 2 plants-11-03410-f002:**
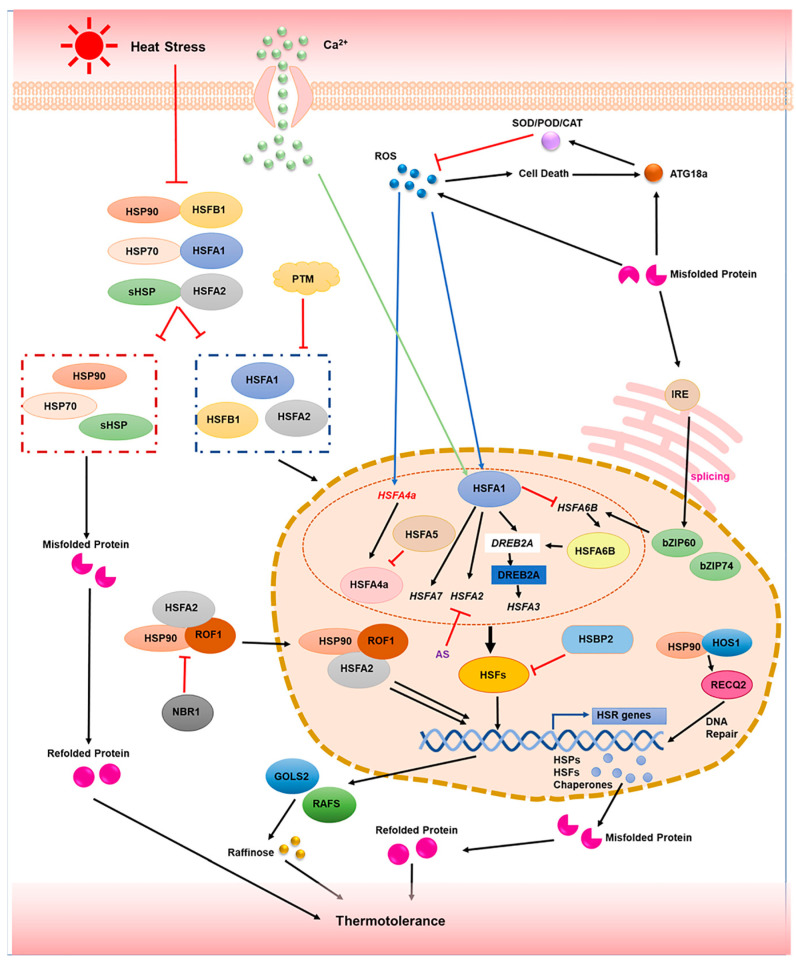
Schematic representation of HSFs-and HSPs-mediated heat stress response (HSR) in plants. The heat signal is perceived by the plasma membrane and induces the production of second messengers such as Ca^2+^ and ROS, which then quickly transmit the heat signal to HSFA1, acting as the master regulating the downstream HSR genes and improving thermotolerance under heat stress.

**Figure 3 plants-11-03410-f003:**
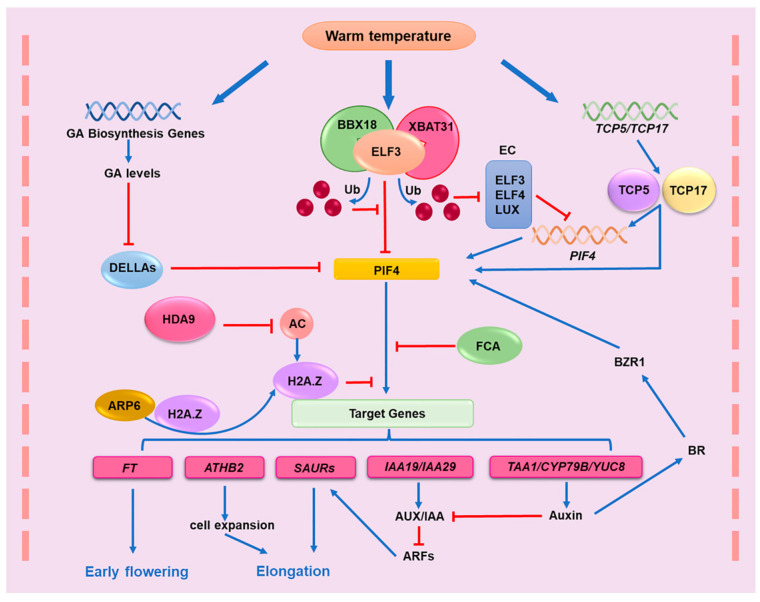
Schematic representation of PIF4-mediated thermomorphogenesis through the regulation of the auxin signaling pathway in plants. Warm temperatures promote PIF4 transcription and protein stability through multiple regulatory mechanisms. PIF4 further positively regulates the auxin response, which promotes thermomorphogenesis in response to moderately elevated ambient temperatures.

**Figure 4 plants-11-03410-f004:**
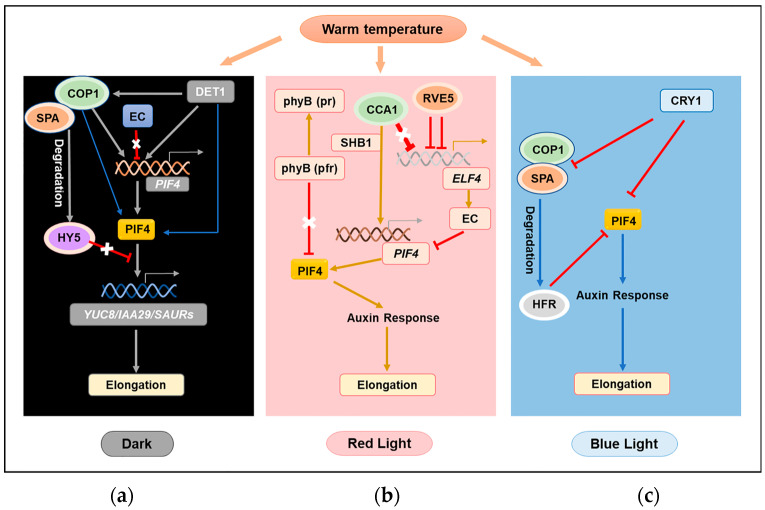
Schematic representation of PIF4-mediated integration of the light signal with the warm temperature signal in plants. PIF4 acts as a cellular “hub” for this integration involving photoreceptors and influences the morphogenic changes by mediating the auxin response in response to a combination of moderately elevated ambient temperature and different light signals. (**a**) PIF4 integrates the darkness signal with the elevated temperature signal and promotes hypocotyl elongation; (**b**) PIF4 integrates the red light signal with the elevated temperature signal and promotes hypocotyl elongation; (**c**) PIF4 integrates the blue light signal with the elevated temperature signal and inhibits hypocotyl elongation.

**Table 1 plants-11-03410-t001:** Previously reported temperature ranges of moderate and extreme heat for different species.

Species	Normal	Moderate Heat	Extreme Heat
*Arabidopsis* [5]	22 °C	28 °C	37 °C
*Zea mays* [6]	25 °C		38 °C
*Oryza sativa* [7]	25 °C	30 °C	35 °C
*Gossypium hirsutum* L. [8]	27.5 °C		36.5 °C
*Malus domestica* [9]	25 °C		48 °C
*Solanum tuberosum* L. [10]	22 °C	30 °C	
*Vitis vinifera* L. [11]	25 °C		35 °C
*Triticum aestivum* L. [12]	24 °C		37 °C
*Marchantia polymorpha* [13]	22 °C		37 °C
*Lilium longiflorum* [14]	22 °C		37 °C
*Solanum lycopersicum* L. [15]	28 °C		42 °C
*Glycine max* L. *Merr.* [16]	28 °C		38 °C
*Petunia hybrida* [17]	26 °C		40 °C

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
