# Peer review of "Regulatory Mechanisms of Heat Stress Response and Thermomorphogenesis in Plants"

_plants, 2022, doi:10.3390/plants11243410_

Round 1

Reviewer 1 Report

This is a good review on mechanisms for plants in response to high temperature and thermomorphogensis in response to moderately elevated temperature. The strengths are this review are the comprehensive analysis of the available literature. Below are some minor editorial suggestions:

Line 24 - Replace 'filed' with 'field'

Line 29-30 - Replace 'predicted' with 'projected'

Line 56 - Replace 'researches' with 'research'

Line 59-60 - Replace 'damages' with 'damage'

LIne 268 - Replace 'signals' with 'signal'

Author Response

Response to Reviewer 1 Comments

Dear Reviewer:

Thank you very much for giving us an opportunity to revise our manuscript, we

appreciate editor and reviewers very much for your critical comments and thoughtful

suggestions on our manuscript entitled “The Regulatory Mechanisms of Heat Stress Response and Thermomorphogenesis in Plants”. These comments are all valuable and very helpful for revising and improving our paper. Below you will find our point-by-point responses to your comments which marked in red in the paper. The main corrections in the paper and the responds to your comments are as following:

Minor revision:Point 1: Line 24 - Replace 'filed' with 'field'.

Response 1: Dear reviewer, thanks for your advice about my typo mistakes. I have replaced 'filed' with 'field' according to your advice.

Point 2: Line 29-30 - Replace 'predicted' with 'projected'.

Response 2: Dear reviewer, thanks for your advice about my typo mistakes. I have replaced 'predicted' with 'projected' according to your advice.

Point 3: Line 56 - Replace 'researches' with 'research'.

Response 3: Dear reviewer, thanks for your advice about my typo mistakes. I have replaced 'researches' with 'research' according to your advice.

Point 4: Line 59-60 - Replace 'damages' with 'damage'.

Response 4: Dear reviewer, thanks for your advice about my typo mistakes. I have replaced 'damages' with 'damage' according to your advice.

Point 5: Line 268 - Replace 'signals' with 'signal'.

Response 5: Dear reviewer, thanks for your advice about my typo mistakes. I have replaced 'signals' with 'signal' according to your advice.

Reviewer 2 Report

The review paper entitled  “The Regulatory Mechanisms of Heat Stress Response and Thermomorphogenesis in Plants” is an interesting overview of the molecular and  regulatory mechanisms involved in plant heat stress response.

The main concerns are the following:

a) the review seems to be mainly based on mechanisms elucidated in Arabidopsis, which is a model plant, and no effort has been made to report the eventual conservation of the same mechanisms also in crops, which are those really important for food security (as mentioned in the introduction). Since there are already similar reviews on Arabidopis the authors should clearly specify the novelty of their work.

b) the authors, in the conclusion, claim that ‘In this review, we present recent advances of research on all these levels of investigation and focus on key integrators HSFs and PIF4, which may help to understand more fully the mechanisms of plants response to heat.’

I think that the literature cited is not really the most recent, during 2021 and 2022 there have been several relevant papers and reviews on the same topic that have not been taken into consideration.

A minor point is about the definition of extreme and moderate stress in terms of temperature ranges, which may vary in a species-specific manner. This information is completely missing.

In order to be really valuable and worth being published the manuscript should be improved as suggested.

Author Response

Response to Reviewer 2 Comments

Dear Reviewer:

Thank you very much for giving us an opportunity to revise our manuscript, we appreciate you very much for your critical comments and thoughtful suggestions on our manuscript entitled “The Regulatory Mechanisms of Heat Stress Response and Thermomorphogenesis in Plants”. These comments are all valuable and very helpful for revising and improving our paper. Below you will find our point-by-point responses to your comments which marked in red in the paper. The main corrections in the paper and the responds to your comments are as following:

Major points:

Point 1: the review seems to be mainly based on mechanisms elucidated in Arabidopsis, which is a model plant, and no effort has been made to report the eventual conservation of the same mechanisms also in crops, which are those really important for food security (as mentioned in the introduction). Since there are already similar reviews on Arabidopsis the authors should clearly specify the novelty of their work.

Response 1: Dear reviewer, thanks for your kind advice. To our best knowledge, although part of the thermal regulatory mechanisms may have been diverged in different species, most of the thermal regulatory mechanisms, particularly in HSF-HSP pathway for mechanistic exploration of HS, has been considered as the eventual conservation in Arabidopsis and also crops [21,37]. And I have also added this explanation (Line 94-98 and Line179-185) and supplemented those relevant literatures on crops such as wheat [12,24,25], rice [26-28], maize [29-31], tomato [15,32], lily [14,33-36] to support the perspectiveson the conservation of HSF-HSP pathway in Arabidopsis and crops under HS.

However, to date, most of the regulatory mechanisms for thermomorphogenesis of plants are mainly studied and interpreted in model plant Arabidopsis, which rarely be further extended to other plant species. Therefore, it could be the focus field in the future to uncover whether there are some conserved regulatory mechanisms for thermomorphogenesis between model plant Arabidopsis and other higher plants, particularly in crops. And I have also added this explanation (Line 188-192) according to your advice.

Point 2:the authors, in the conclusion, claim that ‘In this review, we present recent advances of research on all these levels of investigation and focus on key integrators HSFs and PIF4, which may help to understand more fully the mechanisms of plants response to heat.’I think that the literature cited is not really the most recent, during 2021 and 2022 there have been several relevant papers and reviews on the same topic that have not been taken into consideration.

Response 2: Dear reviewer, thanks for your kind advice. I have supplemented the most recent relevant papers and reviews in the thermal field during 2021 and 2022 in this manuscript, such as the references [13], [24], [31], [32],[34], [36], [37] and [45], [46], [84], [110],[121], [127], which are listed as follows:

  1. Wu, T.; Hoh, K.L.; Boonyaves, K.; Krishnamoorthi, S.; Urano, D. Diversification of heat shock transcription factors expanded thermal stress responses during early plant evolution. Plant Cell. 2022, 34, 3557-3576.
  2. Lin, J.; Song, N.; Liu, D.; Liu, X.; Chu, W.; Li, J.; Chang, S.; Liu, Z.; Chen, Y.; Yang, Q. et al. Histone acetyltransferaseTaHAG1 interacts withTaNACL to promote heat stress tolerance in wheat. Plant Biotechnol. J. 2022, 20, 1645-1647.
  3. Zhao, Y.; Du, H.; Wang, Y.; Wang, H.; Yang, S.; Li, C.; Chen, N.; Yang, H.; Zhang, Y.; Zhu, Y. et al. The calcium-dependent protein kinase ZmCDPK7 functions in heat‐stress tolerance in maize. J. Integr. Plant Biol. 2021, 63, 510-527.
  4. Pierroz, G. Feeling the heat: discovery of a feedback loop regulating thermotolerance in tomato and Arabidopsis. Plant J.2022, 112, 5-6.
  5. Wang, C.; Zhou, Y.; Yang, X.; Zhang, B.; Xu, F.; Wang, Y.; Song, C.; Yi, M.; Ma, N.; Zhou, X. et al. The heat stress transcription factor LlHsfA4 enhanced basic thermotolerance through regulating ROS metabolism in lilies (Lilium Longiflorum). Int. J. Mol. Sci. 2022, 23, 572.
  6. Zhou, Y.; Wang, Y.; Xu, F.; Song, C.; Yang, X.; Zhang, Z.; Yi, M.; Ma, N.; Zhou, X.; He, J. Small HSPs play an important role in crosstalk between HSF-HSP and ROS pathways in heat stress response through transcriptomic analysis in lilies (Lilium longiflorum). BMC Plant Biol. 2022, 22.
  7. Gao, Z.; Zhou, Y.; He, Y. Molecular epigenetic mechanisms for the memory of temperature stresses in plants. J. Genet. Genomics. 2022.
  8. Chen, Z.; Galli, M.; Andrea, G. Mechanisms of temperature-regulated growth and thermotolerance in crop species. Curr. Opin. Plant Biol. 2022, 65, 102134.
  9. Friedrich, T.; Oberkofler, V.; Trindade, I.; Altmann, S.; Brzezinka, K.; Lämke, J.; Gorka, M.; Kappel, C.; Sokolowska, E.; Skirycz, A. et al. Heteromeric HSFA2/HSFA3 complexes drive transcriptional memory after heat stress in Arabidopsis. Nat. Commun. 2021, 12.
  10. Li, N.; Bo, C.; Zhang, Y.; Wang, L. PHYTOCHROME INTERACTING FACTORS PIF4 and PIF5 promote heat stress induced leaf senescence in Arabidopsis. J. Exp. Bot. 2021, 72, 4577-4589.
  11. Qin, W.; Wang, N.; Yin, Q.; Li, H.; Wu, A.; Qin, G. Activation tagging identifies WRKY14 as a repressor of plant thermomorphogenesis in Arabidopsis. Mol. Plant. 2022, 15, 1725-1743.
  12. Perrella, G.; Bäurle, I.; Zanten, M. Epigenetic regulation of thermomorphogenesis and heat stress tolerance. New Phytologist.2022, 234, 1144-1160.
  13. Xu, X.; Yuan, L.; Xie, Q. The circadian clock ticks in plant stress responses. Stress Biology. 2022, 2.

Minor revision:

Point 1: A minor point is about the definition of extreme and moderate stress in terms of temperature ranges, which may vary in a species-specific manner. This information is completely missing.

Response 1: Dear reviewer, thanks for your suggestions. As you mentioned here, different plants may have discrete temperature thresholds in response to extreme and moderate high temperature. Some species are responsive to temperature changes even in the differences of 1°C and other species are not [2]as I have added in Line 30-35 according to your advice. Also, according to your advice, I have added some previously reported temperature ranges of moderate and extreme heat in different species in Table 1.

Table 1. The previously reported temperature ranges of moderate and extreme heat in different species.

Species

Normal

Moderate heat

Extreme heat

Arabidopsis [5]

22°C

28°C

37°C

Zea mays [6]

25°C

38°C

Oryza sativa [7]

25°C

30°C

35°C

Gossypium hirsutum L. [8]

27.5°C

36.5°C

Malus domestica [9]

25°C

48°C

Solanum tuberosum L. [10]

22°C

30°C

Vitisvinifera L. [11]

25°C

35°C

Triticum aestivum L. [12]

24°C

37°C

Marchantia polymorpha [13]

22°C

37°C

Lilium longiflorum [14]

22°C

37°C

Solanum lycopersicum L. [15]

28°C

42°C

Glycine max L. Merr. [16]

28°C

38°C

Petunia hybrida [17]

26°C

40°C

Round 2

Reviewer 2 Report

The authors improved the manuscript and now it is worth being published